# Heterochromatin assembly and transcriptome repression by Set1 in coordination with a class II histone deacetylase

**David R Lorenz, Lauren F Meyer, Patrick J R Grady, Michelle M Meyer, Hugh P Cam\***

Department of Biology, Boston College, Chestnut Hill, United States

**Abstract** Histone modifiers play essential roles in controlling transcription and organizing eukaryotic genomes into functional domains. Here, we show that Set1, the catalytic subunit of the highly conserved Set1C/COMPASS complex responsible for histone H3K4 methylation (H3K4me), behaves as a repressor of the transcriptome largely independent of Set1C and H3K4me in the fission yeast *Schizosaccharomyces pombe*. Intriguingly, while Set1 is enriched at highly expressed and repressed loci, Set1 binding levels do not generally correlate with the levels of transcription. We show that Set1 is recruited by the ATF/CREB homolog Atf1 to heterochromatic loci and promoters of stress-response genes. Moreover, we demonstrate that Set1 coordinates with the class II histone deacetylase Clr3 in heterochromatin assembly at prominent chromosomal landmarks and repression of the transcriptome that includes *Tf2* retrotransposons, noncoding RNAs, and regulators of development and stress-responses. Our study delineates a molecular framework for elucidating the functional links between transcriptome control and chromatin organization.

**\*For correspondence:** hugh.
cam@bc.edu

**Competing interests:** The authors declare that no competing interests exist.

**Reviewing editor**: Ali Shilatifard, Northwestern University Feinberg School of Medicine, United States

## Introduction

The packaging of eukaryotic DNA with histones into chromatin provides ample opportunities for chromatin-modifying factors to exert extensive control over many aspects of genome-based processes (*Kouzarides, 2007*). In particular, enzymes catalyzing the covalent posttranslational modifications of histones are increasingly seen as critical regulators of transcription and the assembly of chromatin into various functional domains (*Henikoff and Shilatifard, 2011*; *Badeaux and Shi, 2013*). Two of the better understood posttranslational modifications of histones are acetylation and methylation. Whereas acetylation of histones by histone acetyltransferases (HATs) is generally associated with gene activation (*Rando and Chang, 2009*), deacetylation of histones by histone deacetylases (HDACs) tends to correlate with gene repression (*Yang and Seto, 2008*). Coordinated activities among HATs result in region-wide hyperacetylated chromatin states, leading to the formation of euchromatin domains supporting active transcription, and conversely, hypoacetylated chromatin states catalyzed by HDACs give rise to heterochromatin domains refractory to transcription (*Grunstein, 1998*; *Grewal and Jia, 2007*). In contrast, histone methylation is associated with either transcriptional activation or repression, and hence, with euchromatin or heterochromatin (*Huisinga et al., 2006*; *Henikoff and Shilatifard, 2011*). Two well-characterized methylation marks occurring on two closely spaced residues near the amino-terminal tail of histone H3 exemplify this pattern (*Grewal and Jia, 2007*). Methylation at lysine 4 of histone H3 (H3K4me) and at lysine 9 (H3K9me) distinguishes euchromatin and heterochromatin, respectively (*Litt et al., 2001*; *Noma et al., 2001*). However, studies from the fission yeast *Schizosaccharomyces pombe* and other systems show that the euchromatic and heterochromatic landscapes are somewhat fluid, with islands of H3K9me transiently assembled within

**eLife digest** Genes can be turned on or off at different times in an organism's life. In humans, yeast and other eukaryotes, this is mainly controlled by the way DNA is packaged with proteins—known as histones—in a structure called chromatin. Genes that are switched on, or only temporarily switched off, are associated with areas of the genome where the chromatin is loosely packed. In contrast, genes that remain switched off for long periods of time are found in regions—known as heterochromatin—where the chromatin is tightly packed.

There are many enzymes that can modify histones to change the structure of chromatin. One enzyme—called Set1—adds a methyl tag to chromatin, which is known to be associated with genes being switched on. However, Lorenz et al. found that Set1 also has other roles in modifying chromatin in the yeast *Schizosaccharomyces pombe.*

The experiments found that Set1 helps to keep genes switched off and that this role is largely independent of its ability to add the methyl tag to chromatin. Set1 is recruited to many sites across the genome by another protein called Atf1, which is involved in the cell's response to environmental stresses. Lorenz et al. believe that this helps to put these genes in a 'poised' off state so that they are ready to be switched on rapidly if needed.

Set1 also works with another protein that removes acetyl tags—which encourage chromatin to be less tightly packed—from histones. Together, both proteins contribute to the assembly of heterochromatin and keep genes involved in development and stress responses switched off when they are not required.

Collectively, these experiments reveal unexpected and important insights into how Set1—which plays critical roles in many aspects of human health including aging and cancer—works in cells.

euchromatin at certain meiotic genes and the 3′ ends of convergent genes (*Cam et al., 2005*; *Huisinga et al., 2006*; *Gullerova and Proudfoot, 2008*; *Zofall et al., 2012*; *Tashiro et al., 2013*). Conversely, the RNA interference (RNAi) and exosome machineries, certain HATs and an active RNA polymerase II (Pol II) have been documented to contribute directly to the assembly of heterochromatin (*Volpe et al., 2002*; *Djupedal et al., 2005*; *Kato et al., 2005*; *Buhler et al., 2007*; *Xhemalce and Kouzarides, 2010*; *Reyes-Turcu et al., 2011*; *Yamanaka et al., 2013*).

These observations point to the potential roles for other chromatin-modifying factors normally associated with euchromatin in heterochromatin assembly. In particular, the *Saccharomyces cerevisiae* homolog of Set1 (KMT2) responsible for H3K4 methylation (H3K4me) has been implicated in transcriptional silencing at a number of genetic elements (*Nislow et al., 1997*; *Krogan et al., 2002*; *Berretta et al., 2008*; *Camblong et al., 2009*; *Kim and Buratowski, 2009*; *van Dijk et al., 2011*). Set1 forms the catalytic engine of a highly conserved chromatin-modifying complex termed Set1C or COMPASS (*Shilatifard, 2012*). Set1C subunits have been shown to be recruited to active Pol II genes and provide the H3K4me signature for the gene-rich euchromatin (*Krogan et al., 2003*; *Ng et al., 2003*). H3K4me can exist in a mono- (H3Kme1), di- (H3K4me2), or tri- (H3K4me3) methylated form (*Kusch, 2012*). The three forms of H3K4me have different distributions, with H3K4me3 and H3K4me2 enriched at gene promoters and gene bodies, respectively (*Cam et al., 2005*; *Pokholok et al., 2005*). H3K4me1 is enriched at the 3′ end of Pol II genes in budding yeast and at enhancers in mammals (*Pokholok et al., 2005*; *Heintzman et al., 2007*). Gene expression profiling analyses ascribe the repressor function of Set1C to H3K4me2 and/or H3K4me3 (*Margaritis et al., 2012*; *Weiner et al., 2012*).

We have recently discovered a role for the *S. pombe* Set1 in the transcriptional repression and genome organization of long terminal repeat *Tf2* retrotransposons and heterochromatic repeats that are dependent and independent of the Set1C complex and H3K4 methylation (*Lorenz et al., 2012*; *Mikheyeva et al., 2014*). In this study, we investigate the regulatory control of the fission yeast transcriptome by Set1 and its associated Set1C subunits. By systematically analyzing the transcriptomes of H3K4me mutants and mutant strains deficient in each of the Set1C subunits, we find that even though loss of H3K4me generally results in derepression, Set1 exerts its repressive function on most of its targets largely independently of the other Set1C subunits and H3K4me. Intriguingly, genome-binding profiles showed that Set1 localization is not linearly correlated with the levels of transcription at its target loci. In addition to localization at active Pol II genes, Set1 localizes to repetitive elements and

repressed loci associated with development and stress-response pathways. Furthermore, we demonstrate that the conserved stress-response ATF/CREB Atf1 transcription factor mediates the recruitment of Set1 and modulates the levels of H3K4me3 at the centromere central cores and ribosomal DNA array. We show that Set1 coordinates with the class II HDAC Clr3 to mediate the assembly of H3K9me-associated heterochromatin and genome-wide repression of diverse transcripts, including *Tf2* retrotransposons, noncoding RNAs, and developmental and stress-response genes. Our study illuminates a surprising cooperation between two histone-modifying enzymes with seemingly opposing activities in imposing genome-wide repression over the transcriptome and organizing the genome into euchromatin and heterochromatin.

## Results

### Set1 behaves as a general repressor largely independent of its H3K4me function and other Set1C subunits

Set1 is the catalytic engine of the Set1C complex that includes seven other subunits (*Roguev et al., 2003*). Except for Shg1, Set1 and six *S. pombe* subunits (Swd1, Swd2, Swd3, Spp1, Ash2, Sdc1) have orthologs in *S. cerevisiae* and humans (*Roguev et al., 2003*; *Shevchenko et al., 2008*; *Shilatifard, 2012*). Loss of individual Set1C complex subunits affects differentially the levels and states of H3K4me in *S. pombe* (*Roguev et al., 2003*; *Mikheyeva et al., 2014*). We performed expression profiling analyses in mutant strains deficient in H3K4me or lacking individual subunits of the Set1C complex. Whereas loss of *set1* resulted in significant derepression of nearly 1000 of ~42,000 tiling microarray probes (average $\log_2$ fold-change vs wild-type >1.5, p < 0.05), H3K4me null mutants *H3K4R* (histone H3 lysine 4 substituted with arginine) or *set1F*[H3K4me−] (H3K4me abolished by Set1 C-terminal FLAG epitope insertion) (*Lorenz et al., 2012*; *Mikheyeva et al., 2014*) affected ~100 probes (*Figure 1A*). Profiling analysis of other Set1C subunits showed a wide range of effects on transcriptional repression, with fewer than 100 probes significantly changed versus wild-type in *ash2Δ* to ~300 in *spp1Δ*. Similar to the other H3K4me mutants, most probes affected in Set1C subunit mutants corresponded to upregulated transcripts, consistent with previous observations in budding yeast showing that loss of H3K4me tends to result in derepression (*Margaritis et al., 2012*; *Weiner et al., 2012*). Importantly, our results show that the major repressive function of Set1 in *S. pombe* occurs largely distinct from H3K4me and the Set1C complex. Variations among Set1C/H3K4me mutants in the proportion of affected probes corresponding to sense, antisense, and intergenic transcripts were also observed (*Figure 1B*), with equal proportions of differentially expressed probes among the three classes of transcripts seen in *set1Δ*, *H3K4R*, and *set1F*[H3K4me−] mutants. Loss of *ash2* primarily resulted in increased sense transcription, and loss of *shg1*, *spp1*, or *swd3* predominantly affected intergenic transcripts.

### Set1C/H3K4me mutants display unique gene expression profiles

Because Set1C/H3K4me mutants displayed varying degrees of transcriptional effects, we performed two-dimensional hierarchical clustering of all differentially expressed probes to gain further insights into their functional relationships. Despite their functions being linked to H3K4me, transcriptional profiles clustered broadly into four distinct groups (*Figure 1C*, upper panel). The loss of *ash2* and *sdc1*, which affected a higher proportion of sense strand probes than in other mutants (*Figure 1B*), shared a subset of upregulated transcripts with significant gene ontology (GO) enrichment for terms common to stress response, including 'response to stress' (p ≈ $10^{-3}$, *ash2Δ*; p ≈ $10^{-18}$, *sdc1Δ*), 'oxidoreductase activity' (p ≈ $10^{-3}$, *ash2Δ*; p ≈ $10^{-11}$, *sdc1Δ* ), and 'generation of precursor metabolites and energy' (p ≈ $10^{-5}$, *ash2Δ*, p ≈ $10^{-3}$, *sdc1Δ*) (*Figure 1—source data 1A*). The profiles of *shg1* and *spp1* mutants formed the second group of predominantly upregulated probes corresponding to diverse intergenic regions and antisense transcripts sharing comparatively weak GO enrichment. The group consisting of *swd1Δ*, *swd2Δ*, *swd3Δ*, *set1F*[H3K4me−], and *H3K4R* mutants included smaller subsets of differentially expressed probes (*Figure 1C*, upper panel), with modestly significant GO enrichment for upregulated transcripts related to stress response and carbohydrate metabolism (*Figure 1—source data 1A*). The profile of *set1Δ* forms its own distinct group, containing a large set of upregulated transcripts including *Tf2* retrotransposons, pericentromeric repeats, and long noncoding RNAs (lncRNAs) that were little affected in the other Set1C and H3K4me mutants (*Figure 1C*, lower panel; *Figure 1—source data 1B*). These results suggest that loss of individual Set1C subunits produces different effects on the transcriptome that could not be fully accounted for by their known contributory roles to H3K4 methylation.

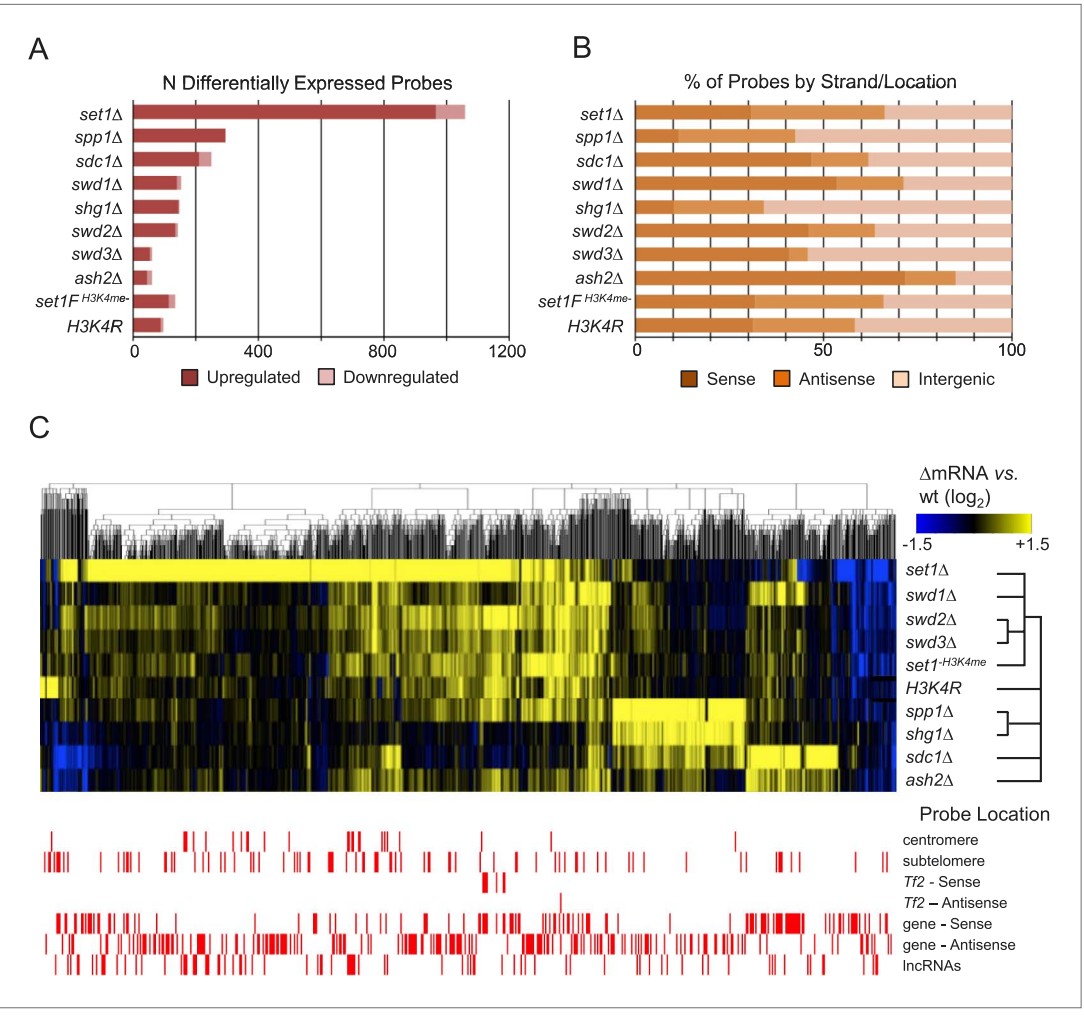

**Figure 1**. Set1/COMPASS subunits act primarily as transcriptional repressors. (**A**) Counts and (**B**) percentage of probes by matching feature strand/position of differentially expressed probes from custom 44,000-probe tiling microarrays. Significantly changed probes were defined as absolute log$_2$ fold-changes ≥ 1.5, false discovery rate (FDR)-adjusted p values <0.05 from duplicate arrays. (**C**) Hierarchical clustering of differentially expressed probes (absolute log$_2$ fold-change vs wild-type ≥1.5, p < 0.05) in Set1C/H3K4me mutant strains. Probes showing significant expression changes in the indicated mutant versus wild-type strains were clustered using the HOPACH algorithm. The bottom panel shows the positions of probes matching repetitive centromeric, subtelomeric (100,000 bp end sequences of all chromosomes), *Tf2* retrotransposons, the sense or antisense strands of annotated protein coding genes, or intergenic long noncoding RNAs (lncRNAs).

The following source data is available for figure 1:

**Source data 1**. Gene ontology (GO) enrichment in Set1C/COMPASS mutant expression profiling microarrays.

**Source data 2**. Comparative analysis of common enriched GO terms in Set1C/COMPASS mutant expression profiling microarrays.

## Set1 localizes to lowly expressed and repressed genes

While H3K4me is known to be enriched at transcriptionally active loci (*Cam et al., 2005*; *Pokholok et al., 2005*), we consistently observed transcriptional derepression in the *set1*Δ mutant at non-active, stress-response genes or heterochromatic repeats. We therefore performed genome-wide mapping of Set1 to gain insights into its repressor function. Consistent with its documented recruitment to active Pol II genes (*Ng et al., 2003*), Set1 is enriched at sites that correspond to highly active Pol II promoters, including those of the housekeeping gene *act1* and the ribosomal protein *rps102* (*Figure 2A*).

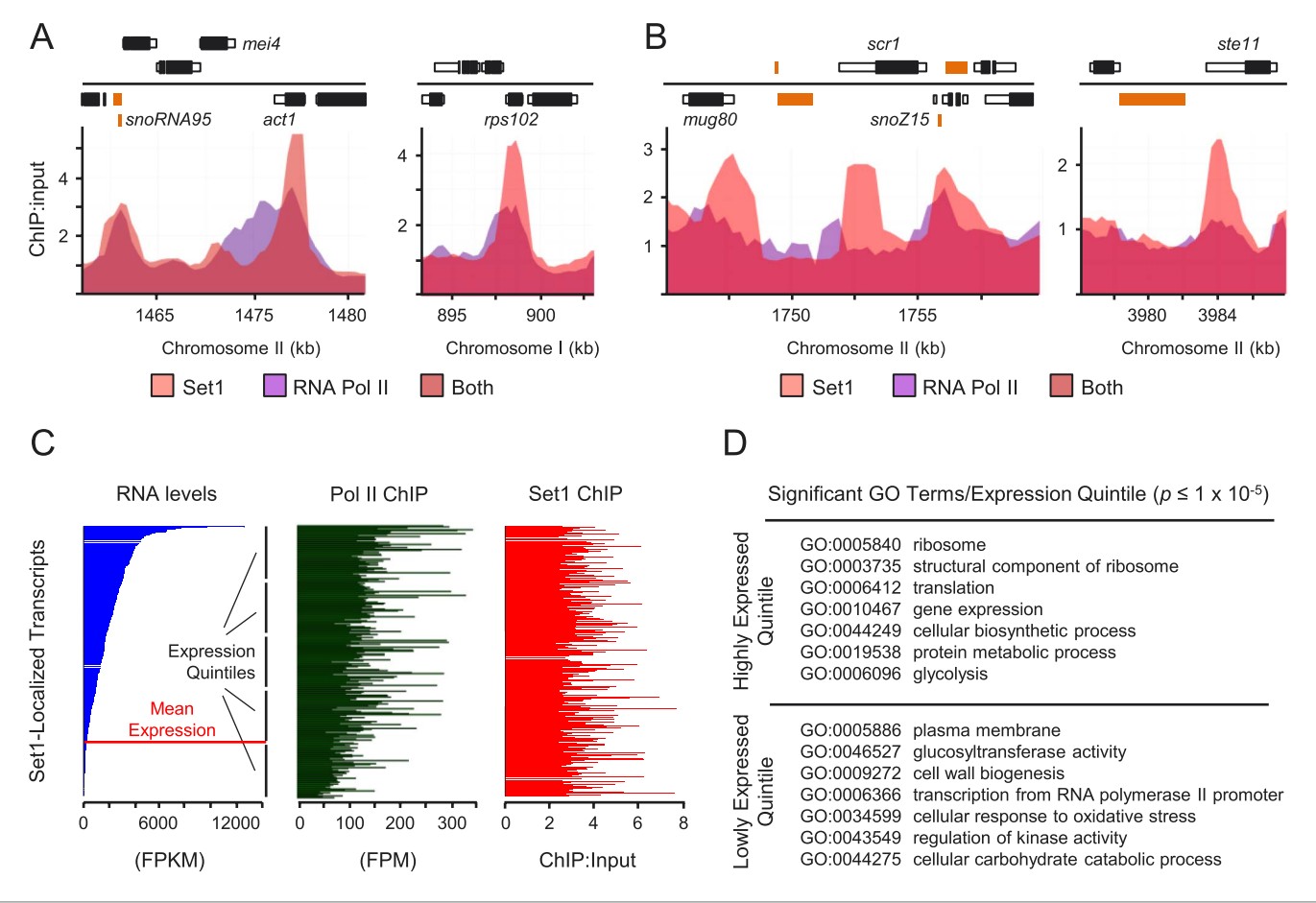

**Figure 2**. Set1 localizes to lowly expressed and repressed loci. (**A** and **B**) Enrichment of Set1 and RNA polymerase II (Pol II) determined by chromatin immunoprecipitation (ChIP)–chip displaying significant Set1 enrichment at highly transcribed genes (**A**) and repressed genes (**B**). Positions of genomic features on forward (top) and reverse strands (bottom), top panel. Black bars denote protein coding gene open reading frames (ORFs); white, associated untranslated regions (UTRs); orange, noncoding RNAs. Pol II ChIP–chip data was derived from *Chen et al. (2008)*. (**C**) Set1 enrichment relative to transcript abundance and Pol II occupancy. Comparisons of RNA-seq expression levels (blue), Pol II ChIP-seq enrichment (green) and Set1 ChIP–chip enrichment (red) at loci showing significant Set1 enrichment (N = 290 transcripts with nonoverlapping annotated features). Processed RNA-Seq FPKM data were obtained from *Rhind et al. (2011)* and Pol II ChIP-seq data from *Zaratiegui et al. (2010)*. The horizontal red line denotes mean expression for all *Schizosaccharomyces pombe* transcripts (*Rhind et al., 2011*). (**D**) Gene ontology (GO) analysis of Set1-bound transcripts by expression level quintile. Representative GO terms were significantly enriched (p ≤ 1 × 10⁻⁵, hypergeometric test) and found exclusively in quintiles of highly expressed (top panel) versus lowly expressed genes (bottom panel). See *Figure 2—source data 1* for a complete list of all significantly enriched GO terms/quintile.

The following source data and figure supplements are available for figure 2:

**Source data 1**. Gene ontology (GO) enrichment of Set1-localized transcripts (ChIP-chip) by target expression level.

**Figure supplement 1**. Set1 localization at active and repressed loci.

**Figure supplement 2**. Distribution of Set1-localized versus all *Schizosaccharomyces pombe* transcripts by absolute expression level.

Surprisingly, despite little enrichment of Pol II at certain lowly expressed genes (e.g., *scr1*) and repressed developmental genes (e.g., *ste11*), noticeable Set1 binding was detected at the promoters of these genes (*Figure 2B*; *Figure 2—figure supplement 1*). Set1 localization at active and repressed targets was not hampered by the loss of H3K4me or its catalytic activity. Indeed, the inability of the *set1F^{H3K4me−}* to methylate H3K4 appears to enhance its association with chromatin. To discern the relationship between Set1 binding and the transcriptional status of its targets, we ranked 290 protein-coding genes with significant Set1 binding (chromatin immunoprecipitation

(ChIP) fold enrichment ≥2 at three or more adjacent probes) according to their expression levels (*Figure 2C*, left panel). While transcript abundance generally correlated with Pol II occupancy levels (*Figure 2C*, middle panel) and 80% of promoter regions enriched for Set1 corresponded to actively transcribed genes (*Figure 2—figure supplement 2*), transcript abundance or Pol II occupancy levels did not linearly correlate with the levels of Set1 binding (*Figure 2C*, right panel). Functional differences between high-abundance and low-abundance Set1-bound genes were assessed by GO analysis of genes rank-ordered by expression levels into quintiles (*Figure 2D*). Whereas highly expressed genes occupied by Set1 were enriched with expected GO terms associated with rapid exponential growth (ribosome, translation, glycolysis), Set1-bound genes with low abundance transcripts (excluding heterochromatic noncoding RNAs due to limited GO annotation) were enriched for terms related to stress response, cell wall and membrane-bound protein biogenesis, and Pol II transcription factor function (*Figure 2—source data 1*). Thus, our results suggest that Set1 localization at chromatin is not solely dependent on active Pol II, and that Set1 localization at lowly expressed or repressed loci might be functionally distinct from its canonical role at active Pol II genes.

## Atf1 mediates recruitment of Set1 at the centromere central cores, rDNA array, and developmental and stress-response genes

A number of low-abundance transcripts shown to be enriched for Set1 in genome-wide binding profiling (e.g., *ste11*) have previously been shown to be targets of the highly conserved ATF/CREB transcription factor Atf1. In addition to localizing to its targets before their activation (*Eshaghi et al., 2010*), which is important for subsequent proper response to environmental stresses (*Chen et al., 2003*), Atf1 contributes to heterochromatic silencing at the silent mating-type locus (*Jia et al., 2004*). We performed genome-wide binding profiling of Atf1 and compared it with that of Set1 to gain insights into the mechanism of Set1 recruitment to chromatin. We observed colocalization of Atf1 and Set1 at centromeric tRNA clusters flanking the euchromatin/heterochromatin boundaries of centromere II and the inner *imr* repeats of the central core (*Figure 3A*, upper panel). Similar colocalization patterns were detected at centromeres I and III (*Figure 3—figure supplement 1*, upper panels). We also detected colocalization of Atf1 and Set1 at the intergenic region of the rDNA and the promoter of the developmental regulator *ste11* (*Figure 3B,C*, upper panel; *Figure 3—figure supplement 2*). We assessed the loss of *atf1* on Set1 activity by mapping distributions of H3K4me3 at these loci in wild-type and *atf1Δ* cells. In wild-type cells, H3K4me3 signals could be detected throughout the centromere central cores and the rDNA array but were little enriched at the *ste11* promoter (*Figure 3A,B, C*; *Figure 3—figure supplement 1*, bottom panels). Loss of *atf1* resulted in a sizeable reduction of H3K4me3 levels throughout the central cores and rDNA array. Moreover, genome-wide analysis identified many loci displaying reduced H3K4me3 in *atf1Δ* compared with wild-type (*Figure 3—source data 1*). The repressed status of the *ste11* gene was not noticeably affected by *atf1Δ* (*Figure 3—figure supplement 4*) and hence has little effect on the status of H3K4me3. However, we noticed that several repressed genes whose promoters are occupied by Atf1 exhibited increased H3K4me3 levels in *atf1Δ* cells (*Figure 3—figure supplement 3*), probably owing to the loss of Atf1-mediated repression.

To determine whether reduced H3K4me3 levels at the centromere central cores and the rDNA array partly reflect the failure of Atf1 to recruit Set1, we assessed Set1 localization at these loci by ChIP. We found that Set1 enrichment at these loci, including the *ste11* gene, was reduced in *atf1Δ* cells (*Figure 3D*). At *ste11*, Atf1 and Set1 appear to act primarily in parallel pathways to keep *ste11* expression repressed, as appreciable upregulation of *ste11* expression was seen only in mutants deficient for both *atf1* and *set1* (*Figure 3—figure supplement 4*). Comparing Atf1 and Set1 localization at the genome scale revealed 217 and 261 distinct bound loci for Atf1 or Set1, respectively, with more than one-third co-occupied by both proteins (p < 0.001, Fisher's exact test) (*Figure 3E*). Collectively, our results suggest that Set1 recruitment to certain repressed loci is mediated in part by Atf1, which in turn is important for proper maintenance of H3K4me levels and, depending on genomic context, transcriptional repression.

## Set1 cooperates with the class II HDAC Clr3 in heterochromatic silencing and the assembly of heterochromatin

To better understand the repressive function of Set1, we sought to identify factors that cooperate with Set1 in heterochromatic silencing. The class II HDAC Clr3 has been shown to contribute to transcriptional silencing of heterochromatin (*Grewal et al., 1998*; *Yamada et al., 2005*) Tf2 retrotransposons (*Hansen et al., 2005*; *Cam et al., 2008*), and stress-response genes (*Lorenz et al., 2012*). These classes of

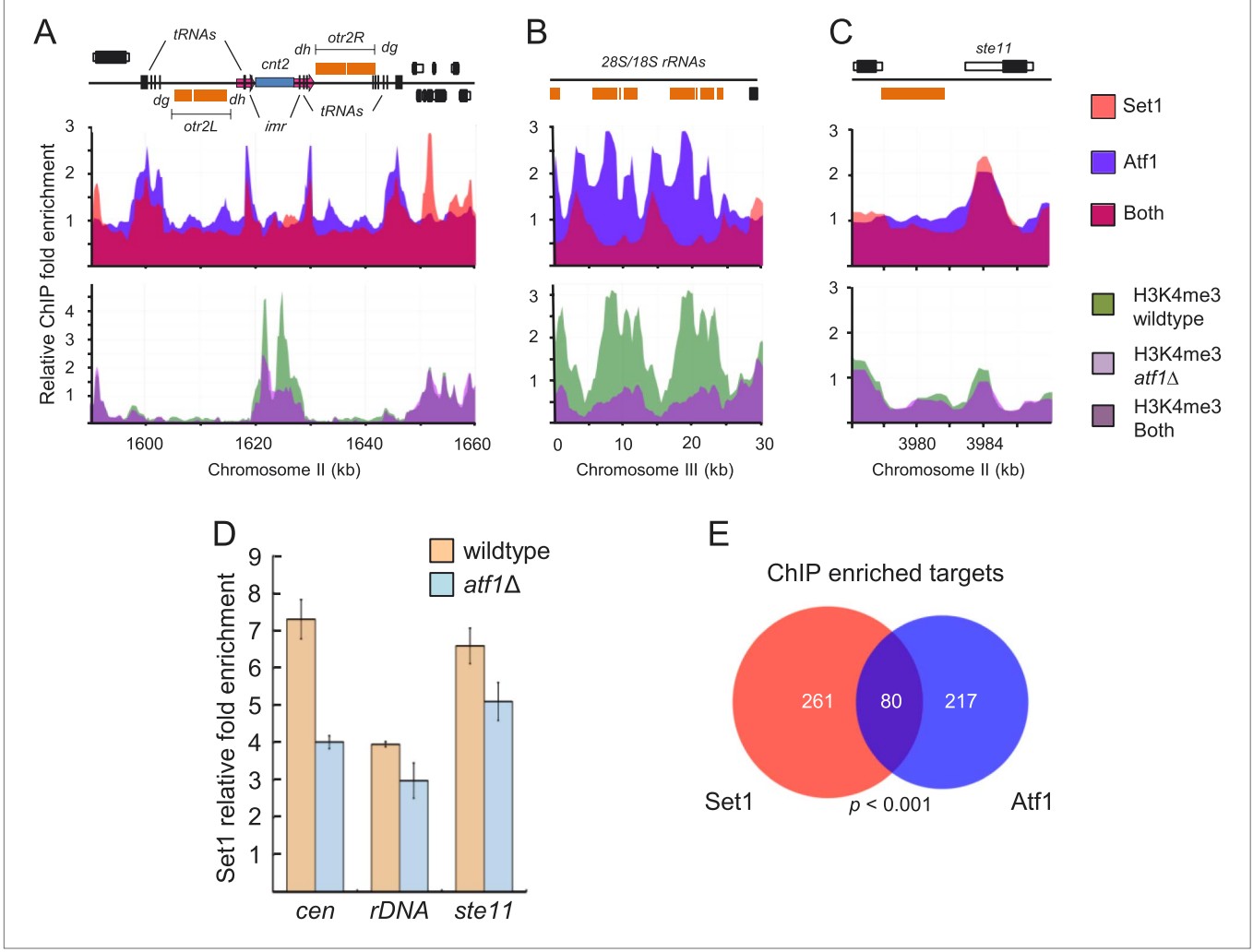

**Figure 3**. Atf1 mediates recruitment of Set1 to centromeres, rDNA, and *ste11* and contributes to H3K4 methylation. (**A**) Colocalization of Atf1 and Set1 (upper panels) at centromere II, (**B**) rDNA array, and (**C**) the promoter of the developmental regulator *ste11*. Enrichment of H3K4me3 (**A–C**, lower panels) and Set1 (**D**) at the aforementioned loci in wild-type and *atf1Δ* cells. Enrichment of Set1, Atf1 and H3K4me3 at indicated loci (**A–C**) was done by chromatin immunoprecipitation (ChIP)–chip. (**E**) Set1 and Atf1 regulate a common set of targets. Venn diagram of Atf1 and Set1 ChIP–chip peaks. Peaks were deemed overlapping if found within 1 kb of each other. The p value was determined by a hypergeometric test with population size $N = 3667$ *Schizosaccharomyces pombe* intergenic regions.

The following source data and figure supplements are available for figure 3:

**Source data 1**. Differential enrichment of H3K4me3 levels in *atf1Δ* vs. wild-type cells.

**Figure supplement 1**. Colocalization of Set1 and Atf1 at centromeres I and III.

**Figure supplement 2**. Enrichment of Atf1 at repressed loci.

**Figure supplement 3**. Atf1 acts as a transcriptional repressor.

**Figure supplement 4**. Derepression of *ste11* in mutants deficient in both *atf1* and *set1*.

genetic elements are also regulated by Set1, suggesting a possible functional link between Clr3 and Set1. To explore this idea, we constructed a mutant strain deficient for both *set1* and *clr3* (*set1Δ clr3Δ*). We observed that in contrast to wild-type or single *set1Δ* or *clr3Δ* mutant strains, a double mutant *set1Δ clr3Δ* strain exhibited a significant synthetic slow-growth phenotype and sensitivity to the tubulin

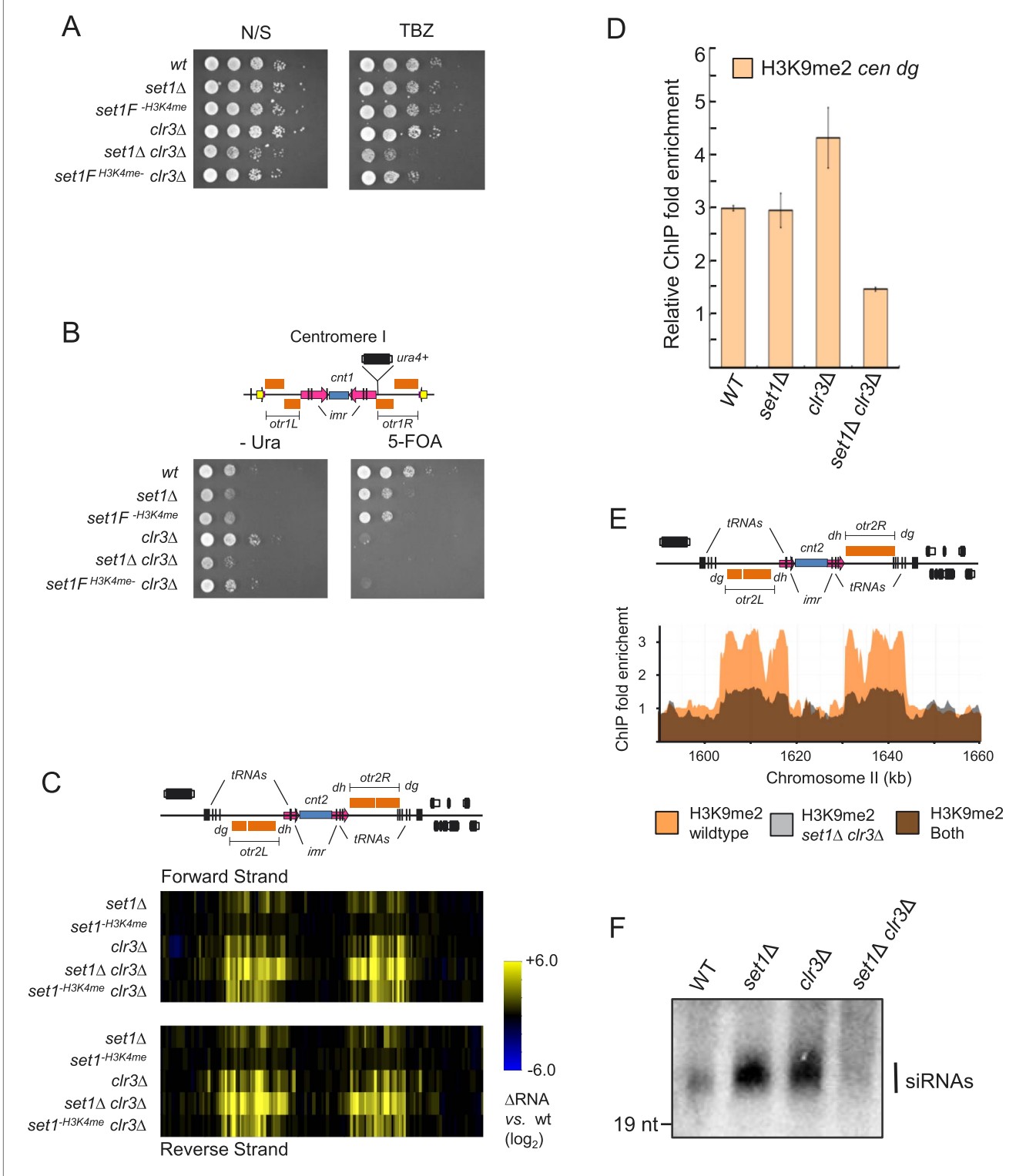

**Figure 4**. Set1 and the class II HDAC Clr3 cooperates in heterochromatic silencing and heterochromatin formation. (**A**) Serial dilution analysis (SDA) of *set1* and *clr3* mutant strains in nonselective (N/S) media or in the presence of the tubulin inhibitor thiabendazole (TBZ), (**B**) uracil minus media (−Ura) or in the presence of the uracil counter selective drug 5-fluoroorotic acid (5-FOA). (**C**) Transcription of forward and reverse strands at centromere II in indicated mutant strains was analyzed by microarrays. (**D**) H3K9 dimethylation (H3K9me2) in strains deficient for *set1* and *clr3* at the pericentromeric *dg* repeat. H3K9me2 enrichment at the *dg* repeat in indicated strains was carried out by chromatin immunoprecipitation (ChIP) and quantified by qPCR. (**E**) H3K9me2
*Figure 4. Continued on next page*

*Figure 4. Continued*

distribution across the entire centromere II in wild-type and *set1Δ clr3Δ* strains. H3K9me2 at centromere II was assayed by ChIP–chip. (**F**) siRNA levels in wild-type, *set1* and *clr3* mutant strains. Detection of siRNAs was carried out by a northern blot using a probe specific for pericentromeric *dg* repeats.

The following figure supplements are available for figure 4:

**Figure supplement 1**. Pol II and Swi6 localization at pericentromeres in *set1* and *clr3* mutants.

**Figure supplement 2**. H3K9me2 defects at centromeres I and III, mating type locus and subtelomeric regions in a strain deficient for both *set1* and *clr3*.

inhibitor thiabendazole (***Figure 4A***), suggesting defects in chromosome segregation. Importantly, the *set1F^H3K4me−^ clr3Δ* double mutant, in which *set1* has no H3K4me activity, exhibited only slight defects. Derepression of a reporter gene inserted within the pericentromeric repeats has been observed in mutants deficient for either *set1* (***Kanoh et al., 2003***) or *clr3* (***Grewal et al., 1998***). We observed additional derepression of the reporter gene in mutants deficient for both *set1* and *clr3* (***Figure 4B***). Defects in heterochromatic silencing result in transcriptional derepression of both the forward and reverse strands of pericentromeric repeats (***Volpe et al., 2002***; ***Moazed, 2011***; ***Alper et al., 2013***). We performed expression analysis using tiling microarrays to assess transcription on both strands in *set1* and *clr3* mutant strains. Modest increases in transcript levels were found on both strands associated with the pericentromeric *dg* and *dh* repeats in single *set1Δ* and *clr3Δ* mutants. However, in the *set1Δ clr3Δ* double mutant, the increase was not only synergistic but occurred throughout the entire pericentromeric region (***Figure 4C***).

Heterochromatin assembly is characterized by the establishment of histone H3 lysine 9 methylation (H3K9me) and HP1/Swi6 proteins bound to H3K9me (***Nakayama et al., 2001***). H3K9me/Swi6 is thought to provide a platform for the recruitment of histone modifiers such as HDACs which could restrict the accessibility of Pol II (***Yamada et al., 2005***). We performed chromatin immunoprecipitation (ChIP) followed by quantitative PCR (qPCR) to monitor the levels of H3K9me, Swi6, and Pol II at the pericentromeric *dg* repeats in the *set1* and *clr3* mutants. Similar to previous observations (***Yamada et al., 2005***), the loss of *clr3* resulted in increased levels of H3K9me2 and Pol II and a decrease in Swi6 enrichment (***Figure 4D***; ***Figure 4—figure supplement 1***). Loss of *set1* resulted in a slight increase of Pol II localization (***Xhemalce and Kouzarides, 2010***) and did not diminish H3K9me2 and Swi6 levels at the *dg* repeats. In contrast, there was a dramatic reduction in the levels of H3K9me2 and Swi6 accompanied by further increase of Pol II occupancy in the double mutant lacking both *set1* and *clr3*. We extended our analysis of H3K9me2 genome-wide and found that H3K9me2 levels in *set1Δ clr3Δ* mutant were reduced across the entire pericentromeric region (***Figure 4E***). H3K9me2 defects in the double mutant were seen at other centromeres and heterochromatin domains, including the silent mating type region and subtelomeres (***Figure 4—figure supplement 2***). The RNAi machinery is known to contribute to the assembly of pericentromeric heterochromatin, in part by acting in *cis* to generate siRNAs (***Volpe et al., 2002***; ***Noma et al., 2004***). We found that whereas loss of *clr3* or *set1* resulted in an increase of siRNAs (***Sugiyama et al., 2007***), the level of siRNAs was dramatically reduced in the double mutant (***Figure 4F***). Thus, our results reveal compensatory mechanisms by Set1 and Clr3 acting in parallel pathways to maintain heterochromatin at major chromosomal landmarks in *S. pombe*.

## Coordinated repression by Set1 and Clr3 on a substantial portion of the *S. pombe* transcriptome

To assess the extent of functional cooperation between Set1 and Clr3 in controlling transcription genome-wide, we performed comparative transcriptome analysis in *set1* and *clr3* mutant cells. While the majority of the differentially expressed probes in the *set1Δ* mutant corresponded to increased expression, loss of *clr3* resulted in 792 probes changing significantly in comparison with wild-type, with approximately equal numbers corresponding to upregulated and downregulated transcripts (***Figure 5A***). Intriguingly, cells lacking both *set1* and *clr3* displayed differential expression of nearly 2900 probes, 2343 of which were upregulated. Loss of H3K4me in a *clr3* null background (*set1F^H3K4me−^ clr3Δ*) did not produce such a drastic change to the transcriptome compared with *set1Δ clr3Δ*, but only reduced the proportion of downregulated transcripts seen in the single *clr3Δ* mutant. Similar proportions of probes corresponding to the sense or antisense strands of known transcripts were differentially expressed across *set1Δ* and *set1Δ clr3Δ* mutants, with the exception of *clr3Δ* cells, which displayed an increased

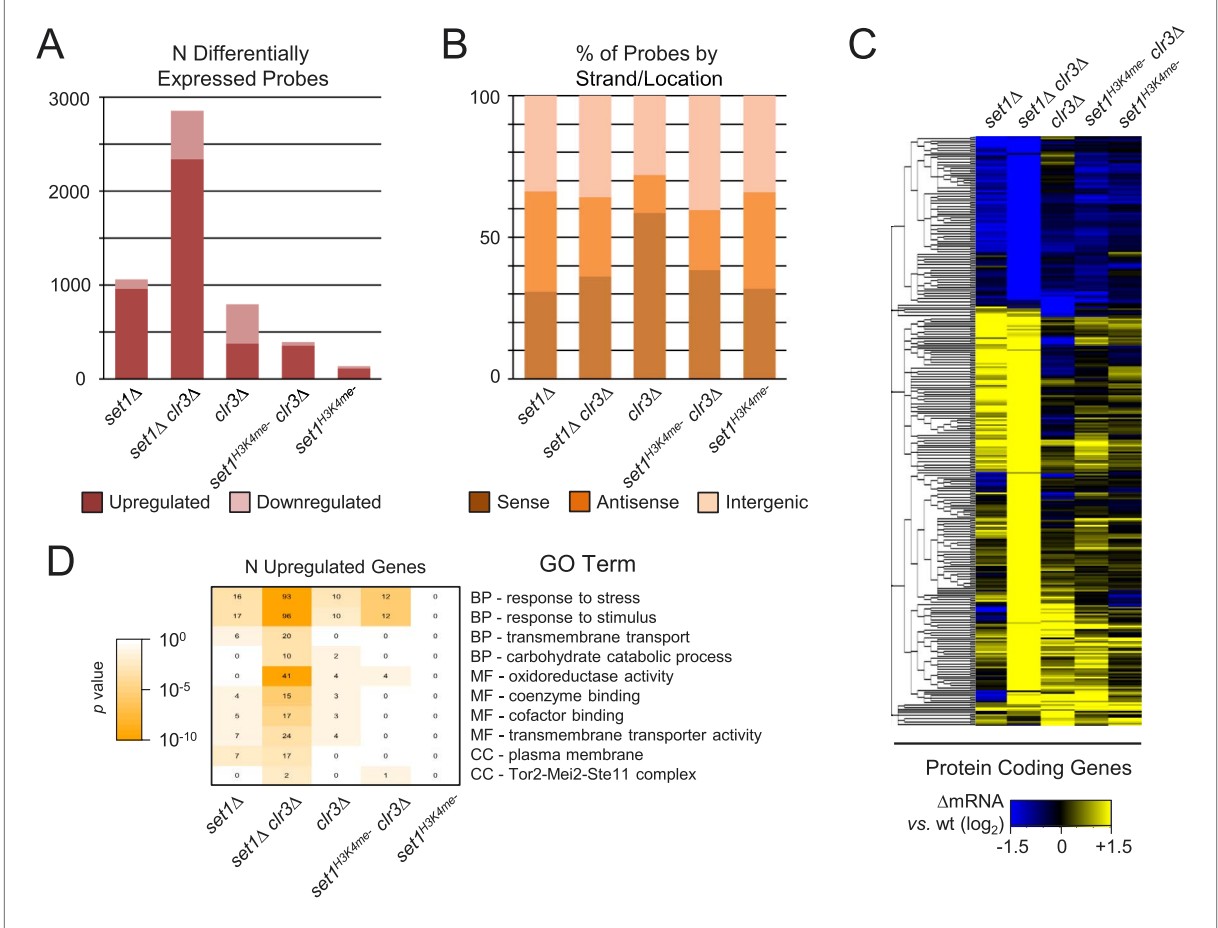

**Figure 5**. Upregulation of a large fraction of the transcriptome in a strain deficient for both *set1* and *clr3*. (**A**) Counts and (**B**) percentage of probes matching feature strand/position in indicated mutant strains were analyzed similarly to *Figure 2A and B*. (**C**) Hierarchical clustering of significantly changed protein coding genes in *set1* and *clr3* mutant gene expression profiles (n = 346). Sense strand probes from two microarray experiments were averaged and clustered as in *Figure 2C*. (**D**) Gene ontology (GO) analysis of upregulated transcripts in *set1* and *clr3* mutant gene expression microarrays. Representative GO terms from biological process ('BP'), molecular function ('MF'), and cellular component ('CC') ontologies displaying most significant enrichment (right panel) and corresponding number of upregulated genes (left panel) in indicated mutant strains; all enriched terms are listed in *Figure 5—source data 1* p values, hypergeometric test.

The following source data and figure supplements are available for figure 5:

**Source data 1**. Gene ontology (GO) term enrichment in *set1/clr3* mutant expression profiling microarrays.

**Figure supplement 1**. Representative genes whose expression requires *set1* and *clr3*.

**Figure supplement 2**. Synergistic upregulation of *Tf2s* and subtelomeric regions in strain deficient for both *set1* and *clr3*.

**Figure supplement 3**. Set1 and Clr3 cooperate to control genes involved in the core environmental stress response.

**Figure supplement 4**. Cooperation between Set1 and Clr3 in development.

proportion of sense strand probes (*Figure 5B*). Hierarchical clustering showed that transcripts downregulated in *set1Δ* tended to be downregulated further in *set1Δ clr3Δ* (*Figure 5—figure supplement 1*), and transcripts that were upregulated in *set1Δ* (i.e., *Tf2s* and subtelomeric regions) were further upregulated in the double mutants (*Figure 5C*; *Figure 5—figure supplement 2*). Most notably, loss of both *set1* and *clr3* resulted in significant expression changes within protein-coding gene regions for a large subset of genes displaying negligible change in individual *set1* or *clr3* mutants (*Figure 5C*).

Upregulated transcripts include well-characterized developmental and stress-response regulatory proteins that include *fbp1*, *mei2* and *ste11* (*Figure 5—figure supplement 3*). Gene ontology analysis suggested that most of the upregulated transcripts in *set1Δ clr3Δ* are associated with stress-response processes that include the Tor2-Mei2-Ste11 pathways (*Figure 5D*; *Figure 5—source data 1*). These pathways are known to be activated during the meiotic development program (*Otsubo and Yamamoto, 2012*). In this regard, we noted that compared with wild-type or single mutant strains, the *set1Δ clr3Δ* double mutant exhibited considerable meiotic defects (*Figure 5—figure supplement 4*). Collectively, our results disclose unexpected coordination between Set1 and Clr3 in ensuring genome-wide repression of the fission yeast transcriptome and proper developmental control.

## Discussion

### Set1C as a repressor complex of the fission yeast transcriptome

Recent transcriptome studies of chromatin mutants in *S. cerevisiae* reveal that loss of *set1* or any of the other four core Set1C subunits (Swd1, Swd3, Bre2/Ash2, Sdc1) produces comparable expression profiles (*Margaritis et al., 2012*). Furthermore, loss of *set1* has only a modest effect on the transcriptome, mainly towards derepression that could fully be accounted by the loss of H3K4me (*Margaritis et al., 2012*; *Weiner et al., 2012*). Similar to these studies, our current study shows that complete loss of H3K4me (i.e., *H3K4R*, *set1F^H3K4me−* mutants) in *S. pombe* has only a slight impact on the transcriptome, with most differentially expressed transcripts upregulated. However, there are important differences. Except for the expression profiles of *H3K4R* and *set1F^H3K4me−* mutants, the profiles among *S. pombe* Set1C subunit mutants are notably disparate, which could not be fully explained by their roles as subunits of Set1C or contributions to H3K4me (*Roguev et al., 2003*). For example, Ash2 and Sdc1 are thought to form heterodimers that together with Swd1 and Swd3 constitute the core of the Set1C complex (*Roguev et al., 2001*; *Dehe et al., 2006*; *Southall et al., 2009*; *Kim et al., 2013*). Yet, while their expression profiles are most similar to each other, there are even differences between them, with the *sdc1* mutant displaying stronger derepression for a subset of genes involved in response to oxidative stress than those seen in the *ash2* mutant (*Figure 1C*). These similarities and differences might reflect their association with other chromatin modifiers such as the Lid2 complex, not present in budding yeast (*Roguev et al., 2003*; *Shevchenko et al., 2008*). Most importantly, the expression profile of *set1Δ* is strikingly different from those of other Set1C/H3K4me mutants, displaying more than eight times the number of upregulated probes relative to those of *swd3* or *H3K4R* mutants. Our findings show that unlike the results reported for *S. cerevisiae*, Set1 in *S. pombe* not only exerts more regulatory influence over the transcriptome, but also mediates its repressive function largely independently of the other Set1C subunits and H3K4 methylation—probably, as a consequence of the uncoupling of Set1 protein stability from H3K4me levels (*Mikheyeva et al., 2014*). Interestingly, *S. pombe* Set1 has been reported as a component of at least two complexes: a large ~1 MDa complex similar in size to that of *S. cerevisiae* Set1C and a smaller complex (~800 kDa) containing a shorter version of Set1 (*Roguev et al., 2003*). Thus, Set1 might mediate its repressive nonH3K4me function via a distinct form of Set1 different from the form associated with the canonical Set1C complex.

### Regulation of repetitive elements, developmental and stress-response loci by Set1 and Atf1

Our study reveals extensive functional interactions across the genome between Set1 and the stress-response transcription factor Atf1 at stress-response genes and major chromosomal landmarks, including the tandem rDNA array and centromeres. At the rDNA array and centromere central cores, Atf1 mediates Set1 recruitment and modulates H3K4me3 levels that might contribute to proper chromatin organization rather than transcriptional repression itself. At loci of stress response and developmental regulators such as *ste11*, Atf1 and Set1 appear to act in parallel pathways that contribute to the repression of *ste11* as loss of both *atf1* and *set1* resulted in significant derepression of *ste11* (*Figure 3—figure supplement 4*). The transcriptional activation of Atf1 is controlled by phosphorylation mediated by the stress-activated mitogen-activated protein kinase (MAPK) Sty1 pathway (*Shiozaki and Russell, 1996*; *Lawrence et al., 2007*). It is likely that co-occupancy of Set1 and Atf1 at the promoters of certain developmental and stress-response regulators not only helps keep these genes in a poised transcriptional off-state, but might also contribute to their rapid transcriptional activation in response to proper developmental or environmental stress signals.

## Functional cooperation between Set1 and Clr3 in heterochromatic silencing and genome-wide repression of the transcriptome

Pol II activity is known to be required for transcriptional silencing and heterochromatin assembly at pericentromeric repeats (*Djupedal et al., 2005*; *Kato et al., 2005*). Other factors associated with active Pol II transcription including components of the Mediator complex have also been shown to contribute to heterochromatin formation (*Oya et al., 2013*). Our study identifies an important role for Set1 in the assembly of heterochromatin domains such as those present at pericentromeres (*Figure 6*). Set1 represses transcription on both the forward and reverse strands of the pericentromeric repeats and cooperates with Clr3 to assemble H3K9me-associated heterochromatin. Importantly, this heterochromatic activity of Set1 appears to be independent of its canonical H3K4me function associated with the Set1C complex, consistent with previous observations for the general lack of H3K4me within H3K9me heterochromatin (*Noma et al., 2001*; *Cam et al., 2005*). Set1-mediated heterochromatin assembly might involve Set1 methylating a nonhistone substrate similar to that of SUV39H1/Clr4 methylating Mlo3, an RNA processing and nuclear export factor that also contributes to RNAi-mediated heterochromatin assembly (*Zhang et al., 2011*). The only known nonhistone target of Set1 is the kinetochore protein DAM1 in *S. cerevisiae* (*Zhang et al., 2005*). However, the *S. pombe dam1* ortholog does not appear to be the target of Set1-mediated heterochromatic silencing as repression of *Tf2* retrotransposons and pericentromeric heterochromatin is maintained in *dam1* mutant cells (Mikheyeva and Cam, unpublished data).

In addition to heterochromatic repeats, a significant fraction of the transcriptome is under repressive control by Set1 and Clr3. Such genome-wide repressive effect strongly suggests that Set1 behaves largely as a bona fide repressor. At developmental and stress-response loci such as *ste11*, Set1 may act in concert with transcription factors, including Atf1 together with Clr3 and other HDACs, to keep the target genes repressed in a steady-state condition. However, unlike heterochromatin, the chromatin states of these loci probably support a transcriptionally poised Pol II and in response to appropriate environmental signals enable Pol II to rapidly upregulate transcription.

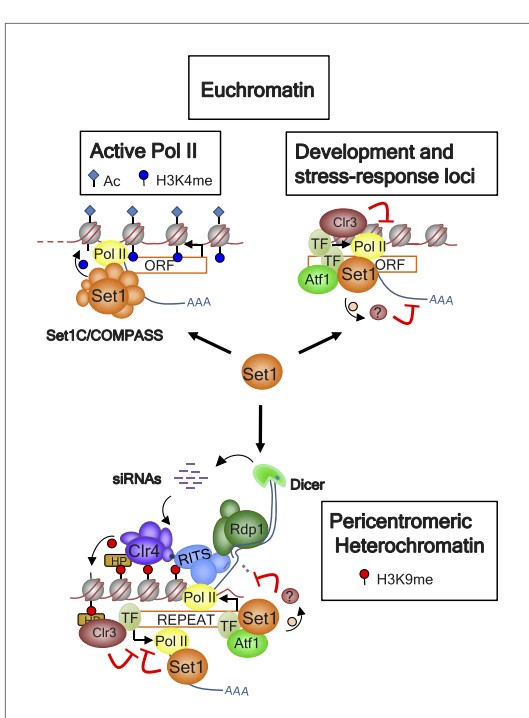

**Figure 6**. Model for Set1 functions at euchromatin and heterochromatin domains. At euchromatin domains, the Set1C/COMPASS complex is recruited to active Pol II genes and provides the H3K4me marks. Set1 is also recruited to certain lowly expressed and repressed genes associated with developmental and stress-response pathways in part by Atf1, other transcription factors (TFs), and probably transcriptionally poised Pol II. Set1 acts in a parallel pathway with the histone deacetylase (HDAC) Clr3 to impose transcriptional repression at these loci. At a heterochromatin domain such as the pericentromeric region, Atf1 and probably other unidentified TFs mediate the recruitment of Set1 to sites enriched for tRNAs known to act as boundary elements. Set1 coordinates with Clr3 in the establishment of SUV39H1/Clr4-mediated H3K9me/HP1 (HP: Swi6 and Chp2) heterochromatin and suppression of bidirectional transcription independently of H3K4me and the other Set1C subunits. Set1-mediated silencing could occur via methylation of nonhistone substrate(s) through the same or different pathways from those of RNAi (i.e., RITS, Rdp1, Dicer) or the exosome (not shown).

## Materials and methods

### Strain Construction

Null mutants of Set1C subunits were constructed using a kanamycin cassette (*Bahler et al., 1998*; *Mikheyeva et al., 2014*). Double mutants were generated by standard genetic cross methods (*Moreno et al., 1991*). Liquid cultures were grown at 30°C in standard rich media supplemented with 75 mg/l adenine (YEA).

## Chromatin immunoprecipitation (ChIP) and ChIP–chip

ChIP assays were performed as previously described (*Lorenz et al., 2012*). ChIP enrichment was quantified by qPCR analysis. ChIP–chip was carried out as previously described using Agilent tiling microarrays (*Cam et al., 2005*). ChIP–chip analysis was performed using the R/Bioconductor *ringo* package (*Toedling et al., 2007*). Preprocessing was carried out by loess normalization. ChIP-enriched regions were defined as three or more adjacent microarray probes with fold-enrichment greater than a two-Gaussian null distribution threshold (greater than twofold enrichment). Between-array analysis of H3K4me3 in wild-type and *atf1Δ* experiments was performed using the *limma* (linear models for microarray data) package after interarray quantile normalization. Antibodies used for ChIP and ChIP–chip assays were anti-FLAG Set1 (M2; Sigma-Aldrich, St. Louis, MO), anti-Atf1 (sc-53172; Santa Cruz Biotechnology, Inc., Dallas, Texas), anti Pol II (ab5408; Abcam, Cambridge, MA), anti-H3K4me3 (07-473; Millipore, Billerica, MA), anti-H3K9me2 (ab1220; Abcam), and anti-Swi6 (*Nakayama et al., 2000*).

## siRNA detection

Small RNAs were purified from 50 ml culture of logarithmically growing cells using the Ambion mir-Vana miRNA/siRNA isolation kit (Life Technologies, Grand Island, NY). Small RNAs (60 μg) were loaded onto a 15% denaturing polyacrylamide gel and run at 300 V until the bromophenol blue dye reached the bottom of the gel (~1.5 hr). Northern transfer was done overnight by capillary blotting in Tris-borate-EDTA buffer at room temperature onto Hybond-N+ membrane (GE Healthcare, Pittsburgh, PA). The membrane was subsequently UV crosslinked twice at 1200 J. Hybridization was carried out in 10 ml ULTRAhyb-Oligo buffer (Life Technologies) at 40°C overnight with a $^{32}$P-labeled RNA probe specific to pericentromeric *dg* repeats. The RNA probe was generated by in vitro transcription using a T7 RNA polymerase system and 50 μCi of [α-$^{32}$P]UTP. Detection of the siRNA signals was carried out using the Storm 820 molecular imager (Molecular Dynamics; GE Healthcare).

## Gene expression profiling

Transcriptional profiling analysis was done as previously described (*Lorenz et al., 2012*). Briefly, RNA was extracted from batch cultures of mid-exponential phase (OD$_{595}$ ~ 0.3–0.6) from mutant and isogenic wild-type strains, reverse-transcribed into cDNA, and labeled with either Alexa Fluor 555 (wild-type sample) or Alexa Fluor 647 (mutant sample) using Superscript Indirect cDNA labeling system (Life Technologies). Equal amounts of labeled cDNA (200–300 ng) from wild-type and mutant samples were mixed and hybridized on a custom 4 × 44k probe Agilent tiling microarray as previously described (*Cam et al., 2005*). For hierarchical clustering using the R/Bioconductor *hopach* package (*van der Laan and Pollard, 2003*), interarray quantile normalization was performed using the *limma* package, and transcripts with more than one differentially expressed probe were averaged. The cosine angle function was used for the clustering distance metric. Gene Ontology (GO) enrichment was performed as previously described (*Lorenz et al., 2012*).

Datasets associated with transcriptional profiling and ChIP–chip experiments in this study can be accessed at the Gene Expression Omnibus under accession number GSE63301.

## Acknowledgements

We thank Grace Kim, Daniel Shams, and Betty Slinger for experimental support, Ke Zhang for the siRNA protocol, Shiv Grewal for the Swi6 antibody, Ee Sin Chen, Irina Mikheyeva, and David Layman for critical reading of the manuscript. Work in the Cam laboratory is supported by the Boston College Wielers Faculty Research Fund and the March of Dimes Basil O' Connor Starter Scholar Research Award.

## Additional information

### Funding

| Funder | Author |
| --- | --- |
| March of Dimes Foundation | Hugh P Cam |
| Boston College | Hugh P Cam |

The funders had no role in study design, data collection and interpretation, or the decision to submit the work for publication.

## Author contributions

DRL, HPC, Conception and design, Acquisition of data, Analysis and interpretation of data, Drafting or revising the article; LFM, PJRG, Acquisition of data, Analysis and interpretation of data, Drafting or revising the article; MMM, Conception and design, Acquisition of data, Drafting or revising the article

# Additional files

## Major datasets

The following dataset was generated:

| Author(s) | Year | Dataset title | Dataset ID and/or URL | Database, license, and accessibility information |
|---|---|---|---|---|
| Lorenz DR, Meyer LF, Cam HP | 2014 | Heterochromatin assembly and transcriptome repression by Set1 in coordination with a class II histone deacetylase | http://www.ncbi.nlm.nih.gov/geo/query/acc.cgi?acc=GSE63301 | Publicly available at NCBI Gene Expression Omnibus. |

**Reporting Standards:** Standard used to collect data: Microarray datasets are deposited at NCBI GEO according to MIAME 2.0 standards.
The following previously published datasets were used:

| Author(s) | Year | Dataset title | Dataset ID and/or URL | Database, license, and accessibility information |
|---|---|---|---|---|
| Zaratiegui M, Vaughn MW, Irvine DV, Goto D, Watt S, Bahler J, Arcangioli B, Martienssen RA | 2010 | CENP-B preserves genome integrity at replication forks paused by retrotransposon LTR | http://www.ncbi.nlm.nih.gov/sra/?term=SRA024710.2 | Publicly available at NCBI Sequence Read Archive. |
| Rhind N, Chen Z, Yassour M, Thompson DA, Haas BJ, Habib N, Wapinski I, Roy S, Lin MF, Heiman DI | 2011 | Comparative functional genomics of the fission yeasts | http://www.ncbi.nlm.nih.gov/sra/?term=SRP005611 | Publicly available at NCBI Sequence Read Archive. |

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
