## [Decision Letter]

Your manuscript titled, “Heterochromatin assembly and transcriptome repression by
Set1 in coordination with a class II histone deacetylase” was reviewed by two
experts in the field and by a member of the Board of Reviewing Editors (BRE). After a
full discussion of the study and the reviews, I am happy to report that the reviewers
and the BRE member found the study of interest to the journal and therefore we are happy
to consider a revised manuscript addressing the following issues:

1) An essential control in the ChIP-ChIP studies of Set1 is the use
*set1* null background in Set1 ChIP-ChIP studies to verify the
significance of the low signals observed. Additionally, you and co-authors need to show
what percentages of Set1 localize at active and repressed regions, respectively?

2) Further, the result of ChIP-ChIP studies in the paper need to be verified with manual
ChIP, and in all ChIP studies proper controls such as untagged strains need to be used
and demonstrated.

3) Co-localization studies do not represent co-recruitment. Please assess Set1 occupancy
in WT and *atf1* null backgrounds and also evaluate H3K4me3 levels
genome-wide in the presence and absence of Atf1.

4) In *S. pombe*, prominent heterochromatin regions include
pericentromeres, subtelomeres, rNDA, and the silent mating type locus. Although Figure 1 has tried to classify the roles of Set1
and COMPASS in different classes of genes, it is still difficult to evaluate if any
classes of genes are preferentially affected by Set1 only or by COMPASS. The authors
need to re-analyze the expression data based on those Set1-occupied genes/regions. Also,
the GO term analyses on Figure 2 did not reveal
any heterochromatin-related terms. The enrichments of Atf1 in heterochromatin regions
are quite clear; however, the color-coding makes the Set1 signals almost invisible in
Figure 3 and Figure 3—figure supplement 2.

5) Set1 is proposed to act in parallel pathway to Clr3 to assemble H3K9me
heterochromatin. However, the exact nature of defects in *clr3set1*
double mutant is not discussed. One possibility is that double mutant is defective in
production of small RNAs that are critical for RNAi-mediated targeting of
heterochromatin. Additional experiments including changes in small RNAs might provide
information about exact cause of the described changes in H3K9me. Moreover, the authors
show widespread upregulation of genes in the double mutant but the biological
significance of these changes has not been addressed. Is the double mutant defective in
stress responses or does it show developmental defects (such as untimely meiosis etc.)?
Inclusion of such results may help connect changes in gene expression to biological
processes.

6) The authors should clarify whether or not this new function of Set1 is truly
independent of its catalytic activity or methylation of its normal substrate (H3K4).
They can either use catalytically dead Set1 (without altering Set1's stability) or
H3K4R mutant (which is preferable).

7) Introduction: in the sentence discussing heterochromatin islands, the authors should
cite a paper by Zofall et al., Science, 335:96, 2012, that discusses dynamic
heterochromatin domains in different parts of the genome. Similarly, the next sentence
discussing RNAi and exosome should include reference to Yamanaka et al., Nature,
493:557, 2013.

---

## [Author Response]

*1) An essential control in the ChIP-ChIP studies of Set1 is the use set1 null
background in Set1 ChIP-ChIP studies to verify the significance of the low signals
observed*.

As pointed out by one of the reviewers, it is often a challenge to detect localization
of enzymatic proteins. However, due to the expected low signals of Set1 at certain loci,
we have now performed additional manual ChIP experiments to verify Set1 localization.
Because we performed ChIP-chip experiment of Set1 using a FLAG antibody (Sigma, M2)
against an epitope tagged Set1, we now include in our manual ChIP verification of Set1
targets a control ChIP experiment against an untagged strain. These results are shown in
Figure 2—figure supplement 1. Our
results are consistent with our recent findings that Set1 localization at active and
repressed loci is generally not dependent of the status of H3K4 methylation (Figure 2 in Mikheyeva et al., PLOS Genetics,
2014).

*Additionally*, *you and co-authors need to show what percentages
of Set1 localize at active and repressed regions, respectively?*

We have now included analysis of the percentage of Set1 ChIP targets at active and
repressed loci. These results are shown in Figure 2—figure supplement 2. Briefly, of the 290 loci whose promoters are
bound by Set1 (ChIP fold enrichment ≥ 2 at 3+ adjacent probes), 80% of those
loci are considered as actively transcribed genes whereas 20% correspond to repressed
genes (RNA level below the mean expression level for logarithmically growing cells;
[49], RNA-seq data).

*2) Further, the result of ChIP-ChIP studies in the paper need to be verified
with manual ChIP, and in all ChIP studies proper controls such as untagged strains
need to be used and demonstrated*.

Please see our reply to question 1. In addition, because the Atf1 ChIP-ChIP experiment
was performed using a commercial antibody specific for endogenous Atf1, we now include a
negative control ChIP experiment using the same antibody in an *atf1*
null strain. This result is present in Figure 3—figure supplement 2 which shows that the enriched signals at Atf1
targets is significantly higher in *atf1*+ compared with
*atf1*Δ, suggesting that the antibody binding is specific to
Atf1. Moreover, our Atf1 ChIP-ChIP results are in high agreement with recently published
ChIP-ChIP data of Atf1 using antibody against an HA epitope of an Atf1-HA tagged strain
(Eshaghi et al., PloS One, 2010).

*3) Co-localization studies do not represent co-recruitment. Please assess Set1
occupancy in WT and* atf1 *null backgrounds and also evaluate H3K4me3
levels genome-wide in the presence and absence of Atf1*.

In addition to showing reduced Set1 enrichment at pericentromeric repeats, the rDNA
array, and ste11 (Figure 3), we have included
additional statistical analyses of the H3K4me3 microarray data to determine all genomic
regions displaying significant changes in H3K4me3 in response to
*atf1*Δ. We found that many loci exhibit reduced levels of H3K4me3
in an *atf1*Δ strain. These data are provided in the new [Supplementary-material SD4-data]. The Results and Methods sections of the manuscript contain additional text
summarizing these results and an explanation of the analytical methods, respectively.
Because Set1 is generally enriched at active genes, it complicates our ability to assess
the effect of Set1 occupancy at Atf1 repressed loci in an *atf1* null
strain compared with wild-type. However, we expect Atf1-mediated repressed loci to
exhibit derepression and hence increased H3K4me3 levels in *atf1*Δ
strain. We now include it in Figure 3—figure supplement 3, which illustrates gene promoters corresponding to Atf1-bound
genes (*fbp1*, *srk1*) that are known to be upregulated in
*atf1*Δ and display increased H3K4me3 levels. A complete list
of all such genomic regions are included in [Supplementary-material SD4-data].

*4) In* S. pombe*, prominent heterochromatin regions include
pericentromeres, subtelomeres, rNDA, and the silent mating type locus.
Although*
Figure 1
*has tried to classify the roles of Set1 and COMPASS in different classes of
genes, it is still difficult to evaluate if any classes of genes are preferentially
affected by Set1 only or by COMPASS. The authors need to re-analyze the expression
data based on those Set1-occupied genes/regions*.

Because most of the upregulated probes are exclusive to individual experiments or groups
of experiments (i.e., probes upregulated in *spp1*Δ and
*shg1*Δ or *sdc1*Δ and
*ash2*Δ differ almost completely with upregulated probes in the
*set1*Δ and other experiments), it is not possible to derive a
common subset of genes likely affected by the COMPASS only. This is probably due to
certain COMPASS subunits (i.e., Set1, Ash2 and Sdc1) having roles outside of the COMPASS
complex that could antagonize the function of COMPASS. However, in order to facilitate a
clearer comparison between the classes of genes affected in the individual
*set1*/COMPASS deletion strains summarized in Figure 1, we have reanalyzed the Gene Ontology enrichment in [Supplementary-material SD1-data]
to encapsulate GO terms with significant enrichment (p < 0.01) by experiment. This
new result is now shown in [Supplementary-material SD2-data]. This analysis replicates the more detailed GO analysis
in [Supplementary-material SD1-data] into a format more amenable to comparing the functional effects of the
various *set1*/COMPASS deletion mutants.

*Also, the GO term analyses on*
Figure 2
*did not reveal any heterochromatin-related terms*.

The most current Gene Ontology mappings provided by pombase.org, the reference genome database
for *S. pombe*, is predominantly limited to annotation for protein coding
genes. The set of mostly noncoding transcripts located within heterochromatin regions
therefore currently has limited GO annotation. Only two *S. pombe*
chromatin regulatory proteins are mapped to the “heterochromatin” GO term
(GO:0000792), vs. transcripts within heterochromatin regions. We have revised the
manuscript to clarify this point.

*The enrichments of Atf1 in heterochromatin regions are quite clear; however, the
color-coding makes the Set1 signals almost invisible in*
Figure 3
*and*
Figure 3—figure supplement 2.

We have reformatted Figure 3 and Figure 3—figure supplement 2 to provide a
better contrast for the enriched ChIP signals of Atf1 and Set1.

*5) Set1 is proposed to act in parallel pathway to Clr3 to assemble H3K9me
heterochromatin. However, the exact nature of defects in* clr3set1
*double mutant is not discussed. One possibility is that double mutant is
defective in production of small RNAs that are critical for RNAi-mediated targeting
of heterochromatin. Additional experiments including changes in small RNAs might
provide information about exact cause of the described changes in
H3K9me*.

We have performed several experiments to further elucidate the nature of functional
interactions between *set1* and *clr3*. First, consistent
with the drastically reduced H3K9me levels in cells null for both *set1*
and *clr3*, the double mutant also exhibits severe depletion of the
HP1/Swi6 proteins at pericentromeric heterochromatin accompanied by increased levels of
Pol II occupancy. These results are now shown in Figure 4—figure supplement 1. We also assessed the levels of siRNAs in
the *set1* and *clr3* mutants. We found that whereas loss
of either *set1* or *clr3* resulted in increased levels of
siRNAs, the siRNA level in the *set1*Δ
*clr3*Δ double mutant was dramatically diminished relative to
wild-type. This is a likely consequence of the failure of the RNAi machinery requiring
H3K9me to act in cis to contribute to heterochromatin assembly (i.e., the RITS complex
tethering to H3K9me via Chp1 to generate siRNAs; Noma et al., Nature Genetics, 2005).
This result is shown in Figure 4.

*Moreover, the authors show widespread upregulation of genes in the double mutant
but the biological significance of these changes has not been addressed. Is the
double mutant defective in stress responses or does it show developmental defects
(such as untimely meiosis etc.)? Inclusion of such results may help connect changes
in gene expression to biological processes*.

We have performed experiments that reveal defects in mating and sporulation in the
*set1*Δ *clr3*Δ double mutant. These new
results are present in Figure 5—figure supplement 4.

*6) The authors should clarify whether or not this new function of Set1 is truly
independent of its catalytic activity or methylation of its normal substrate (H3K4).
They can either use catalytically dead Set1 (without altering Set1's stability)
or H3K4R mutant (which is preferable)*.

We have performed additional experiments that show Set1 localization at active and
repressed loci is not impaired due to the absence of H3K4 methylation
(*set1F*
^*H3K4me-*^) or its catalytic activity
(*set1-SET*Δ). These new results are now shown in Figure 2—figure supplement 1.

*7) Introduction: in the sentence discussing heterochromatin islands, the authors
should cite a paper by Zofall et al., Science, 335:96, 2012, that discusses dynamic
heterochromatin domains in different parts of the genome. Similarly, the next
sentence discussing RNAi and exosome should include reference to Yamanaka et al.,
Nature, 493:557, 2013*.

We have now included these references to the indicated sentences in the revised
manuscript.